# Rumination Detection in Sheep: A Systematic Review of Sensor-Based Approaches

**DOI:** 10.3390/ani13243756

**Published:** 2023-12-05

**Authors:** Stephanie Janet Schneidewind, Mohamed Rabih Al Merestani, Sven Schmidt, Tanja Schmidt, Christa Thöne-Reineke, Mechthild Wiegard

**Affiliations:** 1Institute of Animal Welfare, Animal Behaviour and Laboratory Animal Science, Freie Universität Berlin, 14163 Berlin, Germany; christa.thoene-reineke@fu-berlin.de (C.T.-R.); mechthild.wiegard@fu-berlin.de (M.W.); 2Department of Biosystems Engineering, Albrecht Daniel Thaer Institute for Agriculture, Humboldt University of Berlin, 10117 Berlin, Germany; 3BITSz Electronics GmbH, 08060 Zwickau, Germany; sven.schmidt@bitsz.de; 4Nuvisan ICB GmbH, 13353 Berlin, Germany

**Keywords:** *Ovis aries*, small ruminants, refinement, rumination, animal welfare, sensor, accelerometer

## Abstract

**Simple Summary:**

Monitoring rumination behavior holds great potential as an objective approach for assessing sheep health and well-being. This systematic review provides an overview of the use of sensors to detect rumination in sheep in peer-reviewed research articles published on PubMed, Web of Science, and Livivo databases between the years 2012 and 2023. Additionally, information on their commercial availability is provided. When applicable, the review lists the performance of these sensors in terms of accuracy, sensitivity, precision, and specificity for rumination detection. Furthermore, the challenges and areas for future research were summarized. Initially, 935 articles were retrieved, where only 17 articles fulfilled the predefined inclusion criteria. The findings indicate that sensor-based systems have been used to monitor and analyze rumination behavior in sheep, but research regarding the optimal sensor position and settings (e.g., epoch settings) is not yet conclusive. Notably, none of the commercially available sensors were specifically designed for sheep. There is a need for tailored sensor solutions for sheep. Utilizing such sensors can improve sheep welfare and enhance our understanding of their behavior in various contexts.

**Abstract:**

The use of sensors to analyze behavior in sheep has gained increasing attention in scientific research. This systematic review aims to provide an overview of the sensors developed and used to detect rumination behavior in sheep in scientific research. Moreover, this overview provides details of the sensors that are currently commercially available and describes their suitability for sheep based on the information provided in the literature found. Furthermore, this overview lists the best sensor performances in terms of achieved accuracy, sensitivity, precision, and specificity in rumination detection, detailing, when applicable, the sensor position and epoch settings that were used to achieve the best results. Challenges and areas for future research and development are also identified. A search strategy was implemented in the databases PubMed, Web of Science, and Livivo, yielding a total of 935 articles. After reviewing the summaries of 57 articles remaining following filtration (exclusion) of repeated and unsuitable articles, 17 articles fully met the pre-established criteria (peer-reviewed; published between 2012 and 2023 in English or German; with a particular focus on sensors detecting rumination in sheep) and were included in this review. The guidelines outlined in the PRISMA 2020 methodology were followed. The results indicate that sensor-based systems have been utilized to monitor and analyze rumination behavior, among other behaviors. Notably, none of the sensors identified in this review were specifically designed for sheep. In order to meet the specific needs of sheep, a customized sensor solution is necessary. Additionally, further investigation of the optimal sensor position and epoch settings is necessary. Implications: The utilization of such sensors has significant implications for improving sheep welfare and enhancing our knowledge of their behavior in various contexts.

## 1. Introduction

Animal welfare is a significant subject of public debate, yet determining welfare based on objective parameters poses challenges. It is crucial to approach animal welfare issues using an objective and scientifically credible framework whenever feasible. Monitoring and analyzing the behavior of animals can provide valuable insights into their overall health and welfare. Alterations in behavior can serve as potential indicators of positive and negative welfare states, including stress, disease, or pain [1]. Sheep, like other prey species, mask pain and stress, which can lead to issues such as untreated pain [2]. In the presence of humans, sheep alter their behavior, which impedes welfare assessment. The deployment of sensors offers a non-invasive method to monitor and analyze sheep behavior in real time, without affecting the animals by personal observation or clinical examination. The following behaviors are usually monitored with sensors: grazing, resting, eating, drinking, walking, running, ruminating, lying, and standing. Sensors have been applied in a variety of contexts to categorize and quantify sheep behavior for different purposes. For example, sensors can be used to monitor change in ewe behavior during parturition events (licking, grazing, rumination, walking, standing, and idling) [3,4,5]. Numerous studies have demonstrated the factors that impact the accuracy, precision, recall (also known as sensitivity), F-score, and specificity of behavior classification when using sensors. These are sensor position (e.g., positioned on the collar vs. the neck, mouth, or ear); sensor type; data collection frequency; time window size; feature construction; and algorithm used [6,7,8,9,10]. A sensor that detects rumination reliably will be useful to assess sheep welfare, because a sudden cessation or reduction in rumination over a period of time is an indication of reduced welfare [11]. For cattle, sensors have been effectively used to detect changes in rumination as an indication of impaired welfare, such as painful conditions and diseases (e.g., mastitis, acidosis, and ketosis) [12,13,14], while comparable studies conducted with sheep are lacking. For cattle, a variety of sensors that detect rumination are commercially available [15,16]. Scientific research involving the analysis of behavioral patterns in sheep, including rumination detection, are seeing an increase in the use of sensors [17]. Some of these studies deployed a commercially available sensor designed for humans, others for cattle, and some sensors were custom-made for the sheep deployed in the research project. Although the process of rumination, which includes regurgitating a bolus, chewing the cud by moving the jaw in a circular motion, and then swallowing the masticated cud [18], is the same for all ruminants, the sensing system developed for cattle (or for goats) to detect rumination may not be applicable for sheep due to morphological differences [19].

Multiple systematic reviews on the use of sensors to detect behavior in sheep have been published. However, no systematic review focused specifically on rumination detection itself. Riaboff et al. [20], who conducted a systematic review of accelerometers used on cows, sheep, and goats, found that many models that detect ruminant behavior exhibit poor generalization, which can compromise their commercial use [20]. Few studies have deployed sensors to assess sheep welfare. Ogun et al. [11] used a commercial tri-axial accelerometer ear-tag sensor designed for cattle (eSense Flex, Allflex, Dallas, TX, USA) to study potential welfare indicators for lambs. The authors suggested that using biosensors to monitor the quality of transport and pre-slaughter handling has the potential to improve animal welfare and predict certain meat quality parameters, such as drip loss [18].

This article reviews the use of sensor systems for the specific purpose of ruminant detection in sheep with the aim of providing an overview of the suitability and commercial availability of sensors for ruminant detection in peer-reviewed research projects.

Specifically, we aim to:Identify the sensors that were either tested, used, or described in peer-reviewed articles involving rumination detection in sheep;Specify the commercial availability of these sensors and their suitability for sheep;Provide an overview of the performance of the identified sensors regarding rumination detection in terms of accuracy, sensitivity, precision, and specificity;Outline the challenges, future directions, and ideas for further development identified in these studies;Draw conclusions regarding the extent to which rumination sensors can be used to reliably assess sheep health and welfare.

By summarizing the existing literature on rumination sensors, we aim to provide an up-to-date understanding of the current state of the field, as well as identify areas for future research and development.

## 2. Materials and Methods

To conduct this systematic review on sensors that can detect rumination behavior in sheep, we used three electronic databases: PubMed, Web of Science, and Livivo. Livivo was incorporated in order to retrieve German publications. Our search string included keywords related to feeding behavior, rumination, sheep, and sensors. The following search string was used: “(Feeding behavior OR feeding behaviour OR rumination OR ruminant OR ruminat* OR biting OR eat* OR forag* OR masticat* OR chew* OR ingest* OR digest* OR graz* OR chewing cud OR cud-chewing) AND (sheep OR ovine OR Ovis aries OR ovis OR ewe OR ram OR lamb OR tup OR yeanling OR flock) AND (sensor)”. This research adhered to the guidelines outlined in the PRISMA 2020 methodology. The search was conducted in January 2023, utilizing a search filter to encompass articles published within the past ten years. For Livivo and Web of Science, the filter “2012–2023” was employed. In the case of PubMed, a filter of “1 January 2012–31 December 2023” was utilized, as this search engine requires specific dates for precision. The language criteria were limited to English and German due to the language skills of the research associates involved.

To be included in this review, articles had to meet specific inclusion criteria. Articles had to be in English or German language. Articles had to be published between 2012 and 2023 to ensure that the technical information described is not outdated. Only peer-reviewed articles were selected in order to ensure that the research described was adequately verified before being published. Articles had to describe findings regarding a sensor that monitors rumination in sheep. Articles were included if one of the following topics were covered: classification performance, challenges and/or limitations regarding sensor technology to detect rumination in sheep, and possible future directions for using sensor technology in this respect.

Using the search terms described above, 935 articles were retrieved from the electronic databases. Three additional articles were identified by further reading on the subject using the search engine Google Scholar ©. After removing duplicates, 664 articles were excluded for various reasons, such as being unrelated to rumination or not related to animal science, as detailed in Figure 1. A total of 57 abstracts were read in full and assessed for eligibility by three different research associates, ultimately finding that 17 articles fully met the inclusion criteria. When screening the abstracts, it was not always clear how relevant the article was. In such cases, the full text was read. An administrative authorization was not required, since no animals experienced any form of pain or distress.

The 17 articles were read thoroughly to identify sensors that were tested, utilized, or discussed in peer-reviewed articles related to rumination detection in sheep. The publications were cataloged in a Microsoft Excel spreadsheet (Version 2019), and the subsequent information was documented: details on commercial availability, assessment of suitability for sheep (e.g., any mention of device loss in the article), performance metrics such as accuracy, sensitivity, precision, and specificity, challenges encountered, insights into future directions, ideas for further development as outlined in the studies, and the evaluation of sheep health and welfare.

## 3. Results

The 17 fully read and evaluated articles fell into one of the following three categories: studies evaluating behavioral classification performance (*n* = 8), scientific reviews (*n* = 6), and studies that address a precise scientific research question (*n* = 3). The result-oriented evaluation of the techniques mentioned in the publications could be presented using the criteria below.

### 3.1. Sensors and Devices

#### 3.1.1. Type of Sensors

Eleven studies tested or used wearable sensors to detect rumination behavior (among other behaviors), and the remaining six publications were reviews that described rumination detection using sensors. Each of the subsequent sensor systems described included an accelerometer. Systems deployed to classify chewing patterns without specifically aiming to detect rumination (e.g., to determine differences between biting and chewing during feeding) were not included. In total, eight different systems were used. Details for these systems are briefly described in Table 1. The most commonly used sensor was the ActiGraph sensor (ActiGraph, Pensacola, FL, USA) (*n* = 4), followed by the BEHARUM device (*n* = 2). Two devices were custom-made for the purpose of scientific studies (those used by Mansbridge et al. [8] and Dos Reis et al. [21]). Some studies deployed multiple sensors (Dos Reis et al. [21]; Turner et al. [7]; Ogun et al. [11]).

#### 3.1.2. Classification Methods

A variety of classification methods were used in the studies reviewed (see Table 1). The algorithm employed to classify rumination behavior affects the quality of classification performance. The most common one was random forest. Hu et al. [9] reported that the random forest machine learning method in combination with mixed window sizes significantly improved classification accuracies for all behavior classes (grazing, ruminating, walking, and standing). This was especially true for walking and ruminating. Moreover, the change in the number of features significantly impacted the classification performance of these behaviors. Price et al. [19] found that classifier performance varied between individual sheep, which has been described before by Barwick et al. [23]. This is likely due to different morphologies among sheep, leading to differences in collar/harness fit and sensor placement. To ensure classifiers are useful on a commercial scale, Price et al. [19] recommended that classifiers should be trained on a sufficient number of animals with different characteristics (e.g., in terms of size, age, morphology) and in different environmental conditions. The authors collected training data on a large number of animals (196) over three two-week periods across two lambing seasons to ensure the generality of their algorithm. Furthermore, the authors suggested that future work should focus on the development of devices with real-time classification algorithms [19]. This should be considered in future research studies.

#### 3.1.3. Classification Performances of Rumination Behavior

Table 1 provides an overview of the best published performances regarding rumination detection. The table includes studies that assessed system performance, totaling eight out of seventeen studies. The percentages included present the highest percentages attained in the respective study, which is considered sufficiently representative according to an overall accuracy rating conducted by Barkved [24]. This assessment was further affirmed by Allwright [25], who provided a specific accuracy score for machine learning models in their descriptions. This approach was chosen to provide a clear and concise summary of the best results achieved in different studies. Presenting a comprehensive overview of all results across various papers would have resulted in far too many variables to fit into one clear table in this publication, since many different settings and sensor positions were tested. Therefore, the authors of this systematic review concentrated on the systems with the best performance. The table is arranged alphabetically according to the last names of the primary authors. The following metrics are listed specifically for ruminant detection: accuracy, sensitivity (also called recall), specificity, and precision.

Among the eight publications assessing system performance, the three-axis accelerometer-based BEHARUM device, as reported by Decandia et al. [6], demonstrated both the highest accuracy and specificity for rumination detection. Notably, it achieved an ac-curacy of 94.2% in the collar position at 300 s epochs and a specificity of 98.9% in the neck position at 180 s epochs. In the collar position, the device was placed within a traditional brazen bell without a clapper, suspended by a lightweight leather tie. For the nape position, the device was securely fixed to the neck part of the halter behind the head. However, in this position, false negatives, meaning the misclassification of rumination as grazing or other activities, were high. Furthermore, this system did not significantly outperform the custom-made wearable device based on the Intel^®^ Quark™ (including an accelerometer and a gyroscope sensor), which yielded a specificity of 97% in the collar data and ear position (Mansbridge et al. [8]). The highest sensitivity and precision for rumination were reportedly achieved with this system in collar position using 39 features of specific eating behavioral activities based on random forest [8]. According to the authors, precision and recall, in addition to specificity, give a more adequate representation of the classification performance when the priority is to classify one specific behavior correctly (in this case rumination), as opposed to overall accuracy in classifying different behaviors. For two of these metrics, Mansbridge et al. [8] yielded the highest results. The authors attribute the good classification performance to the use of accelerometer- and gyroscope-based features. The Daily Diary and GPS devices (di Virgilio [22]) on the other hand, which combined data from an animal-attached multi-sensor tag (tri-axial acceleration, tri-axial magnetometry, temperature sensor, and Global Positioning System) with landscape layers from a Geographical Information System, did not successfully distinguish rumination from resting. Instead, resting and rumination were interpreted as one behavior. A sensing system consisting of two tri-axial accelerometer sensors (ActiGraph wGT3X-BT and ActiGraph LLC, Pensacola, FL, USA) [3] placed in a halter on the left side of the face (over the cheek), achieved a concordance percentage between observed and predicted rumination behavior of 95 ± a standard deviation of 10. The concordance percentage indicates how often the two sources of data agree or align with each other.

#### 3.1.4. Commercial Availability of Sensors and Sensor-Applications

The following list names sensors that are commercially available according to the publications included in this systematic review:ActiGraph (wGT3X-BT; ActiGraph, LLC, Pensacola, FL, USA);GENEActiv (Activinsights Ltd., Kimbolton, Cambridgeshire, UK);eSense Flex (Allflex, Dallas, TX, USA);Axivity sensors (Axivity Ltd., Newcastle, UK).

However, none of these sensors were initially designed specifically for sheep.

##### Suitability of Commercially Available Sensors Used on Sheep

This section outlines the suitability of the devices used for sheep based on information described in the articles included in this systematic review (apart from information on their purpose for a different species and battery lifespan). It is important to note that there are further advantages and limitations than those described in the following sections; however, this review focuses solely on those discussed in the articles retrieved through the method of this systematic review.

ActiGraph Sensor:

The ActiGraph sensor was designed to be worn around the wrist, waist, ankle, and thigh of humans [26]. The device is able to hold up to 4.0 GB of raw data. Almasi et al. [27] used ActiGraph sensors embedded into halters and positioned at the left side of the muzzle of 10–11-month-old Merino sheep (45.8 ± 14 kg) to determine the distributions and quantify the variation among animals with respect to the times spent grazing, ruminating, idling, walking, and licking. The authors reported that some of the sensors fell off of the animals during the study, and some of them did not record the necessary information, without specifying the proportion of animals affected. Sohi et al. [3] also described that the study using an ActiGraph sensor on first-cross Merino ewes (Merino × Border Leicester and East Friesian, *n* = 32) was limited by a loss of sensors from ewes and malfunctioning of the sensors. Turner et al. [7], on the other hand, did not mention any difficulties when using such sensors on 30 Merino ewes (18 months of age) positioned under the jaw, over the course of the recording phase (which lasted 7 days). Similarly, Hu et al. [9] attached ActiGraph sensors around the neck of Merino ewes for a period of 48 h with an elasticated strap and did not describe any practical difficulties. This, however, does not ensure that all sensors functioned well at all times.

GENEActiv sensor:

The GENEActiv sensor was designed to measure activity in humans while worn around the wrist [28]. Price et al. [19] attached the sensors to 196 different animals (collar-mounted accelerometers were attached to 76 ewes and harness-mounted accelerometers were attached to 120 lambs for consecutive day periods averaging 10.09 ± 3.35 days across two lambing seasons (September/October 2019 and December 2020). The devices were housed in a water-resistant case along with a rechargeable lithium polymer battery, allowing them to withstand the array of weather conditions experienced by a free-ranging sheep flock. These devices can hold up to 0.5 GB of raw data. A single collar holding the GENEActiv sensor was able to be fitted in under 30 s, while a single harness took approximately 1–2 min and extensive manipulation of lambs was needed. The authors described that collars required no further intervention until removal, whereas some harnesses needed to be adjusted multiple times during the deployment due to the rapid growth of lambs, and therefore some individuals needed to be repeatedly recaptured. Three collars and two harnesses were removed early when animals were removed from the flock due to lamb rejections or health issues, but the majority remained attached for the entire deployment period.

eSense Flex Sensor:

The eSense Flex Sensor was designed to measure motion and rumination activity in cattle (calves and adult). The battery lifespan is 3 years [29]. Caja et al. [30] reported that it was possible to attach the devices to the ear of 20 large-sized sheep (24 months old, 80 kg Biellese suckling ewes) [30]. The authors described that sensor analysis effectively determined significant changes in rumination and motion activity in response to stressful events (manure removal and cleaning of the barn; weaning of the sucking lambs; and shearing of the flock). No ear or sensor problems were reported, such as loss of tag, tearing of ear, breakage, or failure, were detected during the entire experiment. Sensor analysis was described to be effective for determining significant changes in rumination and motion activity as responses to the stressing events. However, marked diurnal variation was reported. The authors note that further research will fully define the utility of using this device under on-farm conditions for sheep behavior monitoring.

Axivity Sensor:

The Axivity sensor was designed to be worn around the wrist of humans. The battery life is 30 days. In a study by Turner et al. [7], tri-axial accelerometer data was recoded to detect rumination in sheep by attaching the sensor to the ear-tag. The authors provided no details about how sensors were attached (e.g., using shrink wrap tubing, as was the case for other studies [31]) and whether there were any difficulties when using the sensors.

##### Scientific Applications of Commercially Available Sensors That Detect Rumination

Three scientific studies used sensors to detect rumination for scientific purposes. This involved rumination detection as a welfare indicator, the quantification of variation in sheep behavior, and estimating heritability of grazing and rumination traits. Table 2 provides more information on these studies. Almasi et al. [27] used ActiGraph (wGT3X-BT; ActiGraph LLC, Pensacola, FL, USA) sensors to study the distributions and variation in behaviors among sheep on a commercial farm with respect to the times spent grazing, ruminating, idling, walking, and licking. In a separate publication, Almasi et al. [4] described that ActiGraph accelerometer sensors can be used to determine grazing and rumination activities of sheep, which has a potential application for breeding strategies.

#### 3.1.5. Challenges Involved in Detecting Rumination Using Sensors

##### Identifying the Ideal Sensor Position for Reliable Rumination Detection

Riaboff et al. [20] noted that ingestive behaviors (grazing, feeding, and ruminating) are best detected using neck-, jaw-, and ear-mounted sensors. For other behaviors, such as motion and postures, other locations yield better classification results. Decandia et al. [6] suggest that placing the sensor on the collar will not disturb the animal, and is thus practical for detecting changes in eating behavior. Price et al. [19] argue that placing the accelerometer on the neck allows for the detection of the majority of primary behaviors performed by sheep; in addition, this arrangement is uncomplicated to deploy and secure enough to avoid equipment loss. Moreover, sensor position impacts the accuracy of the algorithm [8]. Studies have investigated which sensor positions perform best in terms of behavior classification. An overview of the studies that tested various sensor positions can be found in Table 1.

Optimal sensor position depends on the behaviors that are targeted, but also animal welfare aspects must be considered. Caja et al. [30] suggested placing PLF (precision livestock farming) devices in collars for monitoring small ruminant activity to avoid problems of size and excessive weight due to large batteries. Decandia et al. [6] found the accuracy and sensitivity for recording rumination were highest when the accelerometer was placed in the collar position. However, the performance in terms of precision and specificity for ruminating was best when the sensor was positioned on the nape. Nonetheless, misclassification of rumination (as grazing or other activities) was high in the nape position. Mansbridge et al. [8] also found that accuracies for behavioral classification were slightly higher in the collar position than in the ear position, but the differences were almost negligible.

Price et al. [19] noted that sensors attached to the ear (and leg) may not accurately detect jaw movements, such as eating and ruminating. Mansbridge et al. [8], on the other hand, argued that the ear position may be more practical given the possibility of integrating the sensor into an ear-tag. Ogun et al. [11] explicitly reported that there were no issues regarding the ear or the sensor (e.g., tag loss, ear tearing, breakage, failure) during the experimental period, suggesting this location may be commercially viable. Decandia et al. [6] also argued that placing a sensor in a location that yields good classification results (such as in a muzzle, as found by Giovanetti et al. [32]) may be reasonable in a research context, but not viable in a commercial context. Rather, placing the sensors in an ear-tag or a collar is more compatible with conventional husbandry practices. Better rumination detection comes at the cost of classifying other behaviors (e.g., grazing) correctly. To address this, Riaboff et al. [20] proposed adding several accelerometers at different positions on every individual animal and adding a variety of sensors, but also pointed out the impracticality of doing so in the field.

Price et al. [19] found that classifier performance varied among individual sheep. This is likely due to individual differences such as skull morphology, which lead to differences in collar/harness fit.

##### Battery Lifespan

The battery lifespan is a major challenge that limits the utility of sensors able to detect rumination. There is a trade-off between energy consumption and classification accuracy [9]. The battery lifespan of sensors for small ruminants must be shorter compared to those developed for cattle. This is primarily because the batteries used for sensors for small ruminants need to be lightweight to ensure continuous wearability. Sohi et al. [3] reported not being able to use all of the recordings due to battery depletion (in addition to malfunctioning). The authors commented on how an extended battery life will allow more refined training algorithms, which will increase the precision of behavior identification. No effective technology for automatic recharging is currently available [30]. Decandia et al. [6] found that using an epoch setting of 300 s should imply a reduction in data recording and battery consumption. Decandia et al. [10] described that the high battery consumption required to send and receive large data sets could be overcome by undertaking preliminary processing of accelerometer data on the device itself. To achieve this, they suggest optimizing the epoch setting by using larger epoch settings, such as 300 s, because short epoch settings decrease the battery duration. Shorter epoch settings result in more frequent processing and transmission of accelerometer data. This increased frequency of data processing can be energy intensive and result in higher battery consumption, leading to a shorter overall battery lifespan of the device [33]. Decandia et al. [10] conclude that if users intend to record good quality data for longer periods (meaning days or weeks), then 60 and 120 s epochs should be chosen, as these conserve battery energy. However, if users desire higher classification performance and shorter recording periods (one day or less), then the 30 s epoch should be chosen. Thus, the optimal device settings before use must be considered carefully, otherwise behavior classification or battery life may be compromised. An increased frequency limits the battery life (higher sampling rate resulting in higher power consumption). Greater frequency provides greater penetration, which is associated with greater potential risk [30].

##### Economic Costs

Turner et al. [7] describe how using tri-axial accelerometers provides a cost- and power-efficient method to monitor sheep behavior. Brown et al. [34] reported that the tri-axial accelerometer became widespread in research because it is, apart from being small and easy to wear, inexpensive. Thus, it seems like sensors have the potential to optimize production, while reducing costs in the sheep industry. However, Caja et al. [30] explain how the current delay in implementation of precision livestock farming systems for small ruminants can be attributed to the low individual profit and the large number of animals usually kept in this sector. One could hypothesize that the interest in this field is small due to the miniaturization process required, coupled with the higher production costs associated with lower manufacturing numbers. Furthermore, the poor technological infrastructure in many sheep farms is a challenge for the implementation of such technologies. Sheep are often held in mountainous regions, meaning such technologies may not function well under these circumstances. According to Caja et al. [30], estrus detection was the main economic driver for PLF systems in dairy cattle, which is not a high priority for small ruminant farmers. Morgan-Davies et al. [35] described that financial issues, such as equipment costs, are the largest barrier for implementation of PLF systems. The authors concluded that financial aid in purchasing the technology, in addition to training on the use of the equipment, might benefit the uptake of these new technologies. As for the research setting, Dos Reis et al. [21] suggested that the wearable sensor tested (see Table 2) is relatively easy and low cost to construct: batteries (USD 12), microprocessor (USD 11), and nine-axis inertial sensor (USD 8). The authors believe that the system proposed can be constructed and augmented by researchers with ease based on the observation that undergraduate student volunteers typically needed less than an hour of training to be able to independently construct a sensor.

### 3.2. Future Directions for the Use and Further Development of Sensors That Detect Rumunation

#### 3.2.1. Improving Sensors and Classification Performances

Dos Reis et al. [21] concluded that substantial additional work is needed for algorithm development, power source testing, and network optimization to improve sensors that can detect sheep behavior. Algorithm development will improve the accuracy and efficiency of data processing, ensuring that the sensors can effectively analyze and interpret the collected information related to sheep behavior [19]. Optimizing the power source will allow systems to operate consistently over extended periods without frequent interruptions [36]. Network optimization is needed for a robust and efficient network to facilitate data exchange [37]. The increased amount of data generated by a larger sample size provides a better dataset for training and improving machine learning algorithms. Sohi et al. [3], who used a sample size of 165 ewes, stated that a larger sample size with advanced sensors, which have an extended battery life-span, will enable more refined algorithms. Further evaluation of different sensor positions with a larger sample size and more advanced machine learning techniques can help identify behaviors with higher precision [3,8]. Decandia et al. [6] suggested that future studies should aim to evaluate the accuracy of accelerometers in the collar position for detecting health problems and for bite counting estimation. Price et al. [19] noted that future studies could include ruminating (and grazing) behaviors in lambs that are weaned and show rumination behavior. Given that rumination detection in the studies included in this systematic review predominantly focused on adult sheep, future studies should include more lambs. Caja et al. [30] argue that research and farm demonstrations should be considered in order to transfer the available technologies to the agricultural sector. Further research on the ability of sensors to assess animal welfare, including the diurnal patterns of activity, is necessary according to Fan et al. [18].

#### 3.2.2. Investigating Rumination as a Welfare Parameter Using Sensors

Authors agree that automated sensor-based monitoring of sheep has large potential in terms of improving animal welfare [30,38,39]. A major component of good animal welfare is good health. In regard to this, Dos Reis et al. [21] stated that automatic detection of rumination can be used to recognize rumination disorders. The authors note that more reliable data to identify rumination can contribute to improvements in commercially available systems.

## 4. Discussion

Utilizing sensors to reliably detect rumination behavior could enable an objective assessment of sheep health and welfare. Given the limited research on the automated detection of rumination as an objective welfare indicator for sheep, it is worth investigating this topic [25]. This systematic review found that rumination detection using sensors still faces challenges. Sensors can rapidly provide data, but there is still a gap in our understanding of how best to manage and leverage this data to deliver optimal value. A sensor designed specifically for sheep is not yet commercially available. The current commercially available sensors deployed for sheep, which were actually designed to be used in humans or cattle, showed some limitations in their suitability for sheep. Some of the devices used showed malfunctioning, or did not gather data after falling off the animal [3]. From another perspective, the articles did not clearly indicate whether additional work was involved when these devices were fitted to sheep. A device tailored especially for sheep would be promising. Determining the ideal sensor position on the animal, and the appropriate number of sensors, is a challenge. Otherwise, a trade-off between the quality of behavior classification and other factors, such as animal welfare and commercial utility, is possible.

The classification of rumination behavior in respect to different performance metrics (accuracy, precision, sensitivity, and specificity) was good for multiple sensors, but varied for different sensor positions and epoch settings. Finding an optimal combination of sensor position and epoch settings, without compromising the battery lifespan, remains a challenge. Furthermore, economic factors remain a challenge that must be faced in the future [3,30]. Developing an objective, reliable system to monitor the welfare of sheep is necessary, seeing as sheep often mask pain and distress until it reaches a severe level.

Rumination detection as a welfare indicator is understudied. By detecting changes in behavior, sensors can provide early warning signals regarding states associated with negative welfare, such as stress, a subclinical state of a disease, or pain, as well as environment disturbances. However, this has not yet been appropriately investigated for sheep. Studies investigating changes in behavior have not adequately addressed rumination as a key parameter. Fogarty et al. [40] studied how accelerometer-based activity monitoring may be able to identify changes in sheep activity associated with clinical presentations of Haemonchus contortus infections (grazing/walking and standing), but excluded rumination activity from any further analysis due to infrequent observations of the activity. Similarly, Barwick et al. [41] used a tri-axial accelerometer to predict lameness in sheep, but did not include rumination activity as one of the studied parameters. However, regarding rumination behavior on its own may not provide enough information on animal health and welfare [18]. Lying, eating, and drinking behavior are also important behavioral parameters for welfare [42]. This should also be investigated thoroughly using sensors in future research.

The studies reviewed in this systematic review revealed that when examining other behaviors (e.g., grazing), adjusting the sensor position and epoch settings yielded different results regarding classification performance. For example, Decandia et al. [6] found that placing the sensor in the collar position showed the best performance for grazing activity overall, with the exception of the sensitivity metric, which led to better results when placing the sensor in the mouth position. The performance varied also according to epoch settings. Mansbridge et al. [8] reported that overall accuracies of 91% for ear and 92% for collar data were obtained with a random forest classification model. Di Virgilio [22] reported the following classification accuracies for each behavior: 93% for grazing, 87% for searching, 97% for fast walking, 79% for vigilance, and 75% for resting. Results of the studies reviewed here do not allow final conclusions about which combination of attachment position, epoch setting, and classification algorithm will yield reliable results for rumination detection. In order to be able to objectively assess the welfare of the sheep, a system that reliably detects rumination, feeding, and resting is therefore required.

The authors of this systematic review are well aware that the use of only three databases and two relevant languages (English and German) could limit a general assessment of the research conducted worldwide on the use of sensors to detect and evaluate rumination behavior in sheep. This most likely meant that some relevant studies were not found; for example, three relevant articles were not found in the three databases, but were discussed in this article anyway. This, however, is a general limitation of systematic reviews [43]. Some relevant aspects were not thoroughly discussed in the articles retrieved through this systematic review, such as how the use of deep learning approaches has the potential to improve classification performance. This was only addressed in one publication (Turner et al. [7]). For instance, challenges stemming from data scarcity or variations in sensor positions can be effectively addressed through the application of transfer learning techniques, as discussed by Mao et al. [44], Kleanthous et al. [45], and Ahn et al. [46].

In any case, future studies should definitely investigate the use of rumination sensors for detection of sheep pain and welfare assessment. For this, the use of multiple sensors may be beneficial (e.g., the use of accelerometer- and gyroscope-based features). In the commercial setting, adding sensors at different positions may be impractical, but in the research context, adding several sensors in different positions may be a reliable welfare indicator. Research has demonstrated that the proportion of time sheep spend standing while ruminating is a reliable welfare indicator [47,48]. Thus, two sensors could be used to classify (1) standing vs. lying behavior, and (2) rumination behavior. Additionally, future studies should use a larger sample size, also including adult sheep, to draw more generalizable conclusions on rumination as a welfare indicator in production systems and during stressful events.

## 5. Conclusions

This systematic review provided an overview of existing sensors that can detect rumination in sheep, outlining their application and performance according to 17 scientific studies. A sensor developed specifically for rumination detection in sheep is not yet commercially available. Systems have been used to monitor and analyze rumination (among other behaviors), but the ideal sensor position and epoch settings are worthy of further investigation in future studies. Closely monitoring rumination behavior with a sensor may be an objective, reliable tool for assessing the health and welfare of individual sheep across various contexts. This, too, warrants further investigation in future studies.

## Figures and Tables

**Figure 1 animals-13-03756-f001:**
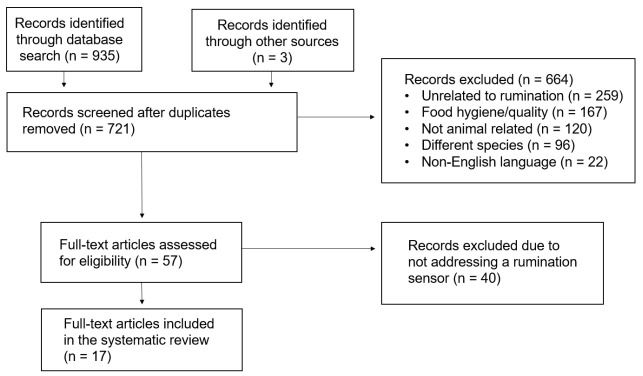
Selection process flow diagram for the literature on the detection of rumination using sensors included in this systematic review based on the PRISMA 2020 guidelines.

**Table 1 animals-13-03756-t001:** An overview of sensor-based rumination detection in sheep: device types, testing environments, animals used, sensor positions and classification metrics.

Reference	Name/Type of Sensor Device	System in Which the Device Was Tested	N, Breed, Sex, Age, Weight, Body Condition Score	Epoch Settings/Window Sizes	Behaviors Classified	Attachment Positions Tested	Method(s) for Classification	Rumination Detection			
								Highest Accuracy Reported	Highest Sensitivity Reported	Highest Specificity Reported	Highest Precision Reported
Decandia et al. (2018) [10]	BEHARUM device (includes a three-axial accelerometer sensor and a force sensor)	Bonassai experimental farm of the agricultural research agency of Sardinia	48 mature lactating Sarda dairy sheep	Epoch settings tested: 5 s, 10 s, 30 s, 60 s, 120 s, 180 s, 300 s	Grazing, ruminating, other activities	Under the lower jaw	CDA and DA	At 30 s (90.0%)	At 120 s (82.2%)	At 30 s (94.7%)	At 30 s (88.1%)
Decandia et al. (2021) [6]	BEHARUM device, which includes a tri-axial accelerometer sensor, inserted in a micro-electromechanical compact system (MEMS)	Experimental farm of the agricultural research agency of Sardinia (grazing system)	3 Sarda ewes, 3.5 ± 0.8 years old, 43.5 ± 1.5 kg, BCS 2.5 ± 0.2	Epochs tested: 5 s, 10 s, 30 s, 60 s, 120 s, 180 s, 300 s.	Grazing, ruminating and other activities	Mouth, nape, collar	Multivariate DA	In collar position at 300 s (94.2%)	In collar position at 300 s (75.0%)	In nape position at 180 s (98.9%)	In nape position at 180 s (81.8%)
di Virgilio (2018) [22]	Daily Diary and GPS devices (CatLog-B, Perthold Engineering) to combine data from an animal-attached multi-sensor tag (tri-axial acceleration, tri-axial magnetometry, temperature sensor and Global Positioning System) with landscape layers from a Geographical Information System	Fortín Chacabuco ranch	3 Merino		Grazing, searching, fast walking, vigilance, and resting	One device was attached to the back of the sheep’s head DD, and on the other one to the neck, attached to the GPS collar	Decision trees for each behavior in the training data set using the “Behavior Building’’ tool from Daily Diary Multi-Trace software (http://www.wildbytetechnologies.com/)	Rumination could not be distinguished from resting, but classification accuracy for resting was 75% in DD position			
Hu et al. (2020) [9]	Tri-axial microelectromechanical systems (MEMS) accelerometers attached to neck collars (ActiGraph wGT3X-BT, Pensacola, FL, USA)	Square mixed sward pasture paddock (70 m × 70 m)	17 Merino ewes	Window sizes tested: 1 s, 2 s, 5 s, 10 s and 15 s	Grazing, ruminating, walking, standing	Devices were attached around the neck of the animals with an elasticated strap	three ML approaches, RF, SVM and LDA	F1-Score best with window size at 2 s and RF method			
Mansbridge et al. (2018) [8]	Custom-made wearable device based on the Intel^®^ Quark™ SE microcontroller C1000 including an accelerometer and a gyroscope sensor	When recordings were taking place, sheep were kept in a rectangular, 0.3-acre field with a 179.3 m perimeter	6 sheep in total, (3 Texel cross, 1 Suffolk cross and 2 Mule), 18 months–4 years old, BCS 2.5 to 4	7-s sample window	Grazing, non-eating behavior, ruminating	Ear and collar	ML algorithms: RF, SVM, kNN, and Adaboost	F-Score using 39 features of specific eating behavioral activities based on RF was slightly higher for collar data (89%) than for ear data (88%)	Recall using 39 features of specific eating behavioral activities based on RF was slightly higher for collar data (87%) than for ear data (86%)	Collar data and ear data both 97% using 39 features of specific eating behavioral activities based on RF	In collar position (92%) using 39 features of specific eating behavioral activities based on RF
Price et al. (2022) [19]	GENEActiv (Activinsights Ltd., Kimbolton, Cambridgeshire, UK) accelerometer-based sensors (wrist-worn devices designed to measure activity in humans)	Commercial sheep farm located in Devon, UK that houses approximately 120 Poll Dorset ewes	196 Poll Dorset sheep (76 ewes and 120 lambs)		Ewes: ruminating, walking; lambs: walking/running, suckling, Both: standing, lying, inactive	Collar-mounted accelerometers (ewes) detected rumination	RF	F-Score for collar position on ewes (only value reported): 76.1%	Sensitivity/Recall for collar position on ewes (only value reported): 77.2%	For collar position on ewes (only value reported): 89.2%	For collar position on ewes 75.0%
Sohi et al. (2022) [3]	(ActiGraph wGT3X-BT; ActiGraph LLC, Pensacola, FL, USA) tri-axial accelerometer sensors	Commercial farm	32 first-cross Merino ewes (Merino × Border Leicester and East Friesian)		Licking, grazing, rumination, walking, and idling	Halters (placed on the left side of the face)		Concordance (Percentage agreement) between observed and predicted rumination behavior: 95 ± 10			
Turner et al. (2022) [7]	ActiGraph sensors (ActiGraph, Pensacola, FL, USA) and ear mounted Axivity sensors (Axivity Ltd., Newcastle, UK)	Muresk Institute Farm	30 Merino ewes, 8 months old	10 s epoch	Sitting, standing, walking, grazing, and ruminating	Jaw and the ear mounted	RF, Long Short-Term Memory, and Bidirectional LSTM	Weighted average F1-score best at RF Baseline (0.84)	Weighted average Recall best at RF Baseline (0.86)		Weighted average precision best with Synthetic Minority Oversampling Techniques (0.83)

Acronyms: Canonical discriminant analysis (CDA), discriminant analysis (DA), machine learning (ML), random forest (RF), Support Vector Machine (SVM), linear discriminant analysis (LDA), k nearest neighbor (kNN), and adaptive boosting (Adaboost).

**Table 2 animals-13-03756-t002:** An overview of the scientific applications of commercially available sensors that detect rumination.

Reference	Name/Type of Sensor Device	N, Sex Age, Weight, Lactation Stage of Sheep	System in Which the Device Was Used	Country	Aim Regarding Rumination Detection	Main Findings Regarding the Detection of Rumination	Attachment Positions Tested
Almasi et al. (2022a) [27]	ActiGraph (wGT3X-BT; ActiGraph, LLC, Pensacola, FL, USA) sensors	147 (male = 67, female = 80) Merino lambs at 10–11 months of age	Commercial farm	Australia	To determine the distributions and quantify the variation among animals with respect to the times spent grazing, ruminating, idling, walking, and licking.	The proportion of each hour spent ruminating varied between 5 and 30 min/h in female sheep whereas male sheep spent as much as 20 min/h in the morning after sunrise. The mean amount ± see of time male sheep spent rumination was 464 ± 3.0 min/day, whereas female sheep spent 399 ± 2.0 min/day.	attached to the left side of the sheep’s muzzle
Ogun et al. (2022) [11]	The commercial ear-tag sensor (eSense Flex, Allflex, Dallas, TX, USA) had been previously tested for use in sheep (Caja et al., 2020 [30]) and were active PLF devices containing a 3-axial accelerometer designed for measuring rumination and motion activity in cattle (calves and adult).	12 Biellese lambs (four females and eight males) and 10 Sambucana lambs (three females and seven males)	Transport and pre-slaughter management	Italy	Precision livestock farming (PLF) technologies were implemented, including accelerometer and rumination activity ear-tag sensors, as potential welfare indicators during transportation and pre-slaughter	Lambs with lower rumination and/or lower total activity were found to have lower drip loss indicating reduced meat quality.	
Almasi et al. (2022b) [4]	ActiGraph (wGT3X-BT; ActiGraph, LLC, Pensacola, FL, USA) accelerometer sensor	147 Merino sheep with the average liveweight of 45.8 ± 14 kg (mean ± S.D.) from 3rd to 29th of May 2020. The ram (*n* = 67) and hogget (*n* = 80)	Commercial farm	Australia	To estimate: (1) the repeatability of grazing and rumination activities between days and during the whole experiment; and (2) the heritability of grazing and rumination activities	Sensor technology and support vector machine method can be applied to determine grazing and rumination activities of sheep with potential application for breeding strategies	To the left side of the muzzle
Dos Reis et al. (2020) [21]	Espressif ESP-32-WROOM-32 microprocessor with Wi-Fi and Bluetooth communication, a generic MPU92/50 motion sensor which contains a three-axis accelerometer, three-axis magnetometer, and a three-axis gyroscope, and a 5-V rechargeable lithium-ion battery.	6 housed adult crossbred Suffolk × Dorset wethers, with an average weight of 70 ± 5 kg (mean ± SD)	Smithfield Farm, Virginia Polytechnic Institute and State University, Blacksburg, VA	USA	To showcase an open-source, microprocessor-based sensor created for the purpose of monitoring and distinguishing various behaviors exhibited by adult wethers	The sensor is able to discern animal behaviors using sensed data (*p* < 0.001). While significant further efforts are required for refining algorithms, testing power sources, and optimizing network functionality, this open-source platform emerges as a promising approach for conducting research on wearable sensors in a broadly applicable manner.	Deployed on a neck collar

## Data Availability

Not applicable.

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
