# Peer review of "Rumination Detection in Sheep: A Systematic Review of Sensor-Based Approaches"

_animals, 2023, doi:10.3390/ani13243756_

Round 1
Reviewer 1 Report
Comments and Suggestions for Authors
- line 24. Senstence on aniamsl masking pain...not sure on the relevance of this for rumination
- lines 186-191 seem contradictory. Please review the information included here
- table 1 and table 2 need to be redone to convey the message you are after. Table 2 label is missing. Both of these tables are confusing and can be structured in a clearer way.
- lines 351-353. As the epoch setting is often a post event analaysis selection, I would question the impact that a longer epoch has on battery life. Do you mean sampling frequency? Please clarify and provide further justification if you mean epoch length.
- line 446. There is a reference missing [X]
Comments on the Quality of English Language
English language is good.
Author Response
Dear Sir or Madam,
We would like to thank you for your thoughtful review and your valuable feedback regarding the article entitled “Rumination Detection in Sheep: A Systematic Review of Sensor-Based Approaches”.
Your comments helped identify areas for improvement, and thus we sincerely appreciate the time and effort you dedicated to reviewing our work. We are committed to making the necessary modifications to enhance the clarity and impact of our article. Our responses to your comments are as follows:
Point 1, concerning line 24: “Sentence on animals masking pain...not sure on the relevance of this for rumination”
Response 1: The authors understand that this may not seem relevant to the readers immediately. We deleted the following text from the abstract: “Sheep are known to mask signs of pain and stress, especially in the presence of an observer, making it challenging to assess their health and welfare accurately. Monitoring rumination behaviour using sensors, which involves the automatic detection of the regurgitation and rechewing of previously consumed feed, holds promise as an objective and reliable tool for evaluating the well-being of individual sheep.”
Point 2, concerning lines 186-191: “seem contradictory. Please review the information included here”
Response 2: We examined the sources upon which the statements were founded and adjusted the text to enhance its clarity. The authors are confident that the revised text effectively communicates the intended message: “Among the eight publications assessing system performance, the three-axis accelerometer-based BEHARUM device, as reported by Decandia et al. [14], demonstrated both the highest accuracy and specificity for rumination detection. Notably, it achieved an ac-curacy of 94.2% in the collar position at 300-second epochs and a specificity of 98.9% in the neck position at 180-second epochs. In the collar position, the device was placed within a traditional brazen bell without a clapper, suspended by a lightweight leather tie. For the nape position, the device was securely fixed to the neck part of the halter behind the head.”
Point 3, concerning table 1 and table 2: “need to be redone to convey the message you are after. Table 2 label is missing. Both of these tables are confusing and can be structured in a clearer way.”
Response 3: The authors believe that the content presented here is important for understanding the best performances achieved for rumination detection and for putting them into perspective (i.e., which settings were used, and understanding that these results were achieved in an experimental farm with adult sheep). In order to clarify the necessity of this table, the following passage was included in Section 3.1.3.: “Table 1 provides an overview of the best published performances regarding rumination detection. The table includes studies that assessed system performance, totaling 8 out of 17 studies. The percentages included present the highest percentages attained in the respective study, which is considered sufficiently representative according to an overall accuracy rating conducted by Barkved (2022). This assessment was further affirmed by Allwright (2022), who provided a specific accuracy score for machine learning models in their descriptions. This approach was chosen to provide a clear and concise summary of the best results achieved in different studies. Presenting a comprehensive overview of all results across various papers would have been far too many number variables to fit into one clear table in this publication, since many different settings and sensor positions were tested. So, the authors concentrated on the systems with the best performance. The table is arranged alphabetically according to the last names of the primary authors.” To enhance visualization, the table should be rotated 90° counter-clockwise and expanded in width. The authors accidentally forgot to include the titles. The title for Table 1 should be: “An overview of sensor-based rumination detection in sheep: device types, testing environments, animals used, sensor positions and classification metrics.” To shorten the text in the table, the classification methods were listed as acronyms. The acronyms for the methods for classification should be listed in a legend in the following manner: “Canonical discriminant analysis (CDA), discriminant analysis (DA), machine learning (ML), Random Forest (RF), Support Vector Machine (SVM) and linear discriminant analysis (LDA), k nearest neighbour (kNN) and adaptive boosting (Adaboost)”. The title of Table 2 should be “An overview of the scientific applications of commercially available sensors which detect rumination.”
Point 4, concerning lines 351-353: “As the epoch setting is often a post event analaysis selection, I would question the impact that a longer epoch has on battery life. Do you mean sampling frequency? Please clarify and provide further justification if you mean epoch length.”
Response 4: Thank you for your critical feedback. The authors understand that this aspect needs to be clarified. The authors do mean epoch setting. Here is a quote from Decandia et al. (Source: Decandia, M., Giovanetti, V., Molle, G., Acciaro, M., Mameli, M., Cabiddu, A., ... & Dimauro, C. (2018). The effect of different time epoch settings on the classification of sheep behaviour using tri-axial accelerometry. Computers and Electronics in Agriculture, 154, 112-119.): “Optimizing the epoch setting, without compromising classification accuracy, could imply a number of advantages. Short epoch settings could increase the labour involved in processing data, deplete the memory device, decrease the battery duration and may also cause erroneous attribution activities during processing. […] optimized longer epoch settings might reduce the memory depletion and increase the battery duration without com-promising the performance of the sensor.“ The following text was included in the manuscript to provide justification that epoch length is correct: “Shorter epoch settings result in more frequent processing and transmission of accelerometer data. This increased frequency of data processing can be energy-intensive and result in higher battery consumption, leading to a shorter overall battery lifespan of the device.”
Point 5, concerning line 446: “There is a reference missing [X]:”
Response 5: Thank you for pointing this out. The following reference was included here: Mattiello, S., Battini, M., De Rosa, G., Napolitano, F., & Dwyer, C. (2019). How can we assess positive welfare in ruminants?. Animals, 9(10), 758.
Kind regards and best wishes from all authors.
Reviewer 2 Report
Comments and Suggestions for Authors
A nice and relevant paper that is informative. Overall I have a comment that the the paper struggles a bit with its focus. I understood that it was dedicated to be able to measure and monitor rumination behaviour of sheep. Of course rumination is related or an indicator for welfare, health, feeding, production etc. But I have the feeling that sometimes these items pop up when they still want to talk on rumination. So on conceptual level and argumentation an additional check by the authors is advised.
A second struggle I see is that the authors want the review to be based on scientific papers. This is well done, but on the other hand their wish to connect it also to solutions that are used in practical commercial situations is sometimes lacking. In the end they only have 4 remaining in this part, but that is not connected to the scientific literature. Especially the link to commercial applications can be tricky. This might be a chicken and egg reasoning.
The challenges and areas for future research lack sometimes a proper reasoning and argumentation. Are they based on finding in the literature an/or on the own expectations of the authors. So, it is not always clear where they are based on.
What surprised me is that most sensors that came back were related to the human sector. I expected much more spinn-off from the rumination sensors in the field of dairy farming. Also in the discussion these were hardly coming back.
L35: Pubmed and Webof Scinece are well known, but I have the impression that Livivo is less known and that it might have a focus on the German language based papers. Maybe an additional explanation might help the readers. Can be done in L116.
L41: .. designed for sheep. Therefore ... There is no direct link. The reasons for customisation for the sheep applications should come from not meeting specific needs and requirements which can be shown by results of the scientific papers.
L65: It is a bit confusing where the 'affect' is related to. It looks like the factor in the same sentence, but that is not the case and they come in the next sentence.
L94-96: Is it needed to add here also the 'commercial use' .
L97-L99: This is the same as L94-96.
L104: I think that the word 'various' should be replace by 'identified'
L112: see earlier remark. If you want to do this then you have to give much more background of the role of rumination in health and welfare, and you have to argument why only health and welfare and you are not discussing feeding and production. It becomes even more difficult if you want to connect rumination to the concept of pain.
L118: Is searching case sensitive. If not then feeding behaviour is double. It might also be that you want to cover both US and UK spelling for behavior and behaviour
L124: give here already the arguments for the language. Later on you give an argument.
L125: Is December 31st 2023 correct. I assume you want to have a 10 years period, other wise I don not see the proper argument to start January 1st 2012.
L138: The inclusion criteria only relate to the scientific papers. In this whole chapter nothing is mentioned about the search and inclusion of the 'commercial' application. This should be added.
L142: Please check numbers in figure. Records screened after duplicates are removed is 718, if you extract 664 excluded papers then not 58 will remain.
L183: the choice is now to present the 8 'best' published performances. Why not present all 17? When you present 8, how do you determine the order in the table. In other words on which criteria and why and how you determine the order. You are aware that they are related to each other. So the reader might be interested in when you think results are acceptable for the sheep sector. This can come back in the discussion and can also be used as argumentation for the improvement and design them specifically for the sheep sector.
L185: add after specificity , and precision.
L214: I am not aware of journal policy but there is no table heading. Also in L293 for table 2.
L216: add of after details
L218-221: This process of selection was not explained yet in chapter 2 and I miss the link with table 1. How can I see in table 1 that they are commercial available.
L271: 'were reported' for this will be correct, but I expect hat in the discussion this should come back. There is a lot of literature and experience with the development of electronic ear tags for identifying sheep and goats.
L294: This whole paragraph is part of the results, but it reads more like a discussion. If it will be results then the identified challenges should be categorised on source. The content of this paragraph is relevant for the paper, but is is more how and where you present it.
L427: see earlier discussion and provide some more background on 'good'.
L484-489: this part of the conclusion is more a wish and speculation. See earlier overall comment. My advise is to restrict the conclusion part to the rumination
L510: I am not aware of the journal policy, but it seems that only first author is mentioned and the rest with 'et al'. I expected full list of authors of each paper.
L517: check this reference, is not complete yet.
Author Response
Dear Sir or Madam,
The authors thank you for your extensive review and on the article titled "Rumination Detection in Sheep: A Systematic Review of Sensor-Based Approaches."
Your comments have been very helpful in pinpointing areas for improvement, and we genuinely appreciate the time and effort you invested in reviewing our work. We are dedicated to implementing the necessary modifications to augment the clarity and impact of our article, and we have edited our manuscript in line with your comments.
Our responses to your comments are outlined below:
General comments: “A nice and relevant paper that is informative. Overall I have a comment that the the paper struggles a bit with its focus. I understood that it was dedicated to be able to measure and monitor rumination behaviour of sheep. Of course rumination is related or an indicator for welfare, health, feeding, production etc. But I have the feeling that sometimes these items pop up when they still want to talk on rumination. So on conceptual level and argumentation an additional check by the authors is advised.
A second struggle I see is that the authors want the review to be based on scientific papers. This is well done, but on the other hand their wish to connect it also to solutions that are used in practical commercial situations is sometimes lacking. In the end they only have 4 remaining in this part, but that is not connected to the scientific literature. Especially the link to commercial applications can be tricky. This might be a chicken and egg reasoning.
The challenges and areas for future research lack sometimes a proper reasoning and argumentation. Are they based on finding in the literature an/or on the own expectations of the authors. So, it is not always clear where they are based on.
What surprised me is that most sensors that came back were related to the human sector. I expected much more spinn-off from the rumination sensors in the field of dairy farming. Also in the discussion these were hardly coming back.“
Response to general comments:
The authors thank you for this feedback. Regarding the comment, “[…] but on the other hand their wish to connect it also to solutions that are used in practical commercial situations is sometimes lacking,” we would like to clarify that this systematic review aimed to identify whether there are any sensors that are commercially available and suitable for sheep. Our search did not specifically target commercial applications; instead, we aimed to gather comprehensive information about rumination sensors for sheep, allowing for an open exploration of the types of articles we would encounter. This review found that sensors which detect rumination in sheep are used in scientific studies. To address your comment that “The challenges and areas for future research lack sometimes a proper reasoning and argumentation.”, we included more explanation in Section 3.2.1. for clarification.
Point 1, concerning L35: Pubmed and Webof Scinece are well known, but I have the impression that Livivo is less known and that it might have a focus on the German language based papers. Maybe an additional explanation might help the readers. Can be done in L116.
Response 1: Thank you for pointing this out. The following sentence was included in the Materials & Methods section: “Livivo was incorporated in order to retrieve German publications.”
Point 2, concerning L41: .. designed for sheep. Therefore ... There is no direct link. The reasons for customisation for the sheep applications should come from not meeting specific needs and requirements which can be shown by results of the scientific papers.
Response 2: This is true. The sentence which your comment refers to was re-written in the following manner: “Notably, none of the sensors identified in this review were specifically designed for sheep. In order to meet the specific needs of sheep, a customized sensor solution is essential. “
Point 3, concerning L65: It is a bit confusing where the 'affect' is related to. It looks like the factor in the same sentence, but that is not the case and they come in the next sentence.
Response 3: The authors understand that this sentence may have been a bit confusing. Therefore, the sentences which this comment refers to were changed in the following manner: “Numerous studies have demonstrated which factors impact the accuracy, precision, recall (also known as sensitivity), F-score, and specificity of behaviour classification when using sensors. These are: sensor position (e.g., positioned on the collar vs. the neck, mouth, or ear); sensor type; data collection frequency; time window size; feature construction and algorithm [14-18].”
Point 4, concerning L94-96: Is it needed to add here also the 'commercial use' .
Response 4: The sentence was modified in the following manner (adding the word “commercial” where it is underlined and in italics): “This article reviews the use of sensor systems for the specific purpose of ruminant detection in sheep with the aim of providing an overview of the suitability and commercial availability of sensors for ruminant detection in peer-reviewed research pro-jects.” “Commercial use” was not added here, because this systematic review assessed the use in scientific research projects and outlined the commercial availability.
Point 5, concerning L97-L99: This is the same as L94-96.
Response 5: Thank you for pointing this out. The following sentences were deleted from the manuscript, seeing as a similar passage comes earlier on in the manuscript: “This article reviews the utilization of sensing systems for the specific purpose of rumination detection in sheep. The aim is to provide an overview of the suitability and availability of sensors which have been deployed to detect rumination in peer-reviewed research projects.”
Point 6, concerning L104: I think that the word 'various' should be replace by 'identified'
Response 6: The word 'various' was replaced with 'identified' accordingly: “3. Provide an overview of the performance of the identified sensors regarding rumination detection in terms of accuracy, sensitivity, precision, and specificity”
Point 7, concerning L112: see earlier remark. If you want to do this then you have to give much more background of the role of rumination in health and welfare, and you have to argument why only health and welfare and you are not discussing feeding and production. It becomes even more difficult if you want to connect rumination to the concept of pain.
Response 7: The sentence “By summarizing the existing literature on rumination sensors, we aim to provide an up-to-date understanding of the state of the field from the perspective of assessing and improving animal welfare, as well as identifying areas for future research and de-velopment.” Was changed to: “By summarizing the existing literature on rumination sensors, we aim to provide an up-to-date understanding of the state of the field, as well as identifying areas for future research and development.”
Point 8, concerning L118: Is searching case sensitive. If not then feeding behaviour is double. It might also be that you want to cover both US and UK spelling for behavior and behavior
Response 8: Thank you for pointing this out. In our search string, we included US and UK spelling of the word “behavio(u)r”. This is the correct search string: Our search string included keywords related to feeding behaviour, rumination, sheep, and sensors. The following search string was used: “(Feeding behavior OR feeding behaviour […]”
The necessary changes were made in the manuscript.
Point 9, concerning L124: give here already the arguments for the language. Later on you give an argument.
Response 9: Thank you for suggesting this. The following sentence was included in the manuscript: “The language criteria were limited to English and German due to the language skills of the research associates involved.”
Point 10, concerning L125: Is December 31st 2023 correct. I assume you want to have a 10 years period, other wise I don not see the proper argument to start January 1st 2012.
Response 10: Yes, this is correct. The following sentences were added to the manuscript: “The search was conducted in January 2023, utilizing a search filter to encompass arti-cles published within the past ten years. For Livivo and Web of Science, the filter "2012 – 2023" was employed. In the case of Pubmed, a filter of "2012/1/1 - 2023/12/31" was utilized, as this search engine requires specific dates for precision.”
Point 11, concerning L138: The inclusion criteria only relate to the scientific papers. In this whole chapter nothing is mentioned about the search and inclusion of the 'commercial' application. This should be added.
Response 11: To address this point, we included details on how we documented the commercial availability of rumination sensors found in the articles included in the systematic review (“The 17 articles were read thoroughly to identify sensors that were tested, utilized, or discussed in peer-reviewed articles related to rumination detection in sheep. The pub-lications were cataloged in a Microsoft Excel spreadsheet (Version 2019), and the sub-sequent information was documented: details on commercial availability, assessment of suitability for sheep (e.g., any mention of device loss in the article), performance metrics such as accuracy, sensitivity, precision, and specificity, challenges encoun-tered, insights into future directions, ideas for further development as outlined in the studies, and the evaluation of sheep health and welfare.”). Our search did not specifically target commercial applications; instead, we aimed to gather comprehensive information about rumination sensors for sheep, allowing for an open exploration of the types of articles we would encounter.
Point 12, concerning L142: Please check numbers in figure. Records screened after duplicates are removed is 718, if you extract 664 excluded papers then not 58 will remain.
Response 12: Thank you very much for pointing out the calculation error we made! We checked our documentation and redid the graphic.
Point 13, concerning L183: the choice is now to present the 8 'best' published performances. Why not present all 17? When you present 8, how do you determine the order in the table. In other words on which criteria and why and how you determine the order. You are aware that they are related to each other. So the reader might be interested in when you think results are acceptable for the sheep sector. This can come back in the discussion and can also be used as argumentation for the improvement and design them specifically for the sheep sector.
Response 13: The authors understand that the passage referred to was a bit confusing. Therefore, the passage was re-written and now cites relevant literature: “Table 1 provides an overview of the best published performances regarding rumination detection. The table includes studies that assessed system performance, totaling 8 out of 17 studies. The percentages included present the highest percentages attained in the respective study, which is considered sufficiently representative according to an overall accuracy rating conducted by Barkved (2022). This assessment was further affirmed by Allwright (2022), who provided a specific accuracy score for machine learning models in their descriptions. This approach was chosen to provide a clear and concise summary of the best results achieved in different studies. Presenting a comprehensive overview of all results across various papers would have been far too many numbers variables to fit into one clear table in this publication, since many different settings and sensor positions were tested. So the authors concentrated on the systems with the best performance. The table is arranged alphabetically according to the last names of the primary authors.”
Point 14, concerning L183: L185: add after specificity , and precision.
Response 14: The grammatical changes were made accordingly.
Point 15, concerning L214: I am not aware of journal policy but there is no table heading. Also in L293 for table 2.
Response 15: The authors propose the following title: “An overview of sensor-based rumination detection in sheep: device types, testing environments, animals used, sensor positions and classification metrics.”
Point 16, concerning L216: add of after details
Response 16: The sentence was adapted to improve clarity as follows: “The following list names which sensors are commercially available according to the publications included in this systematic review.”
Point 17, concerning L218-221: This process of selection was not explained yet in chapter 2 and I miss the link with table 1. How can I see in table 1 that they are commercial available.
Response 17: A passage was added to the Materials & Methods section: “The 17 articles were read thoroughly to identify sensors that were tested, utilized, or dis-cussed in peer-reviewed articles related to rumination detection in sheep. The publications were cataloged in a Microsoft Excel spreadsheet (Version 2019), and the subsequent in-formation was documented: details on commercial availability, assessment of suitability for sheep (e.g., any mention of device loss in the article), performance metrics such as ac-curacy, sensitivity, precision, and specificity, challenges encountered, insights into future directions, ideas for further development as outlined in the studies, and the evaluation of sheep health and welfare.”
Point 18, concerning L271: 'were reported' for this will be correct, but I expect hat in the discussion this should come back. There is a lot of literature and experience with the development of electronic ear tags for identifying sheep and goats.
Response 18: The authors agree that this is true, but this paper focuses on rumination, and the authors believe that discussing the development of electronic ear tags for identifying sheep and goats is too far away from the subject of rumination and would therefore prefer not to include this in the discussion.
Point 19, concerning L294: This whole paragraph is part of the results, but it reads more like a discussion. If it will be results then the identified challenges should be categorised on source. The content of this paragraph is relevant for the paper, but is is more how and where you present it.
Response 19: The authors can understand where this constructive criticism is coming from, and therefore put these two sentences into the discussion section: “Determining the ideal sensor position on the animal, and the appropriate number of sensors, is a challenge. Otherwise a trade-off between the quality of behaviour clas-sification and other factors, such as animal welfare and commercial utility, is possible”. Additionally, another scientific paper was cited.
Point 20, concerning L427: see earlier discussion and provide some more background on 'good'.
Response 20: The authors believe that this point has been sufficiently addressed by the addition of the aforementioned passage (see Response 13).
Point 21, concerning L484-489: this part of the conclusion is more a wish and speculation. See earlier overall comment. My advise is to restrict the conclusion part to the rumination
Response 21: This part of the conclusion was deleted: “A sensor will be beneficial for stressful or painful time-periods, such as: transportation and pre-slaughter; postpartum period; during or after clinical disease; as well as nutrition research and agricultural or biomedical sciences.” And the following was added to the beginning: “This systematic review provided an overview of existing sensors which can detect rumination in sheep, outlining their application and performance according to 17 scientific studies.”
Point 22, concerning L510: I am not aware of the journal policy, but it seems that only first author is mentioned and the rest with 'et al'. I expected full list of authors of each paper.
Response 22: Thank you for point this out – this happened due to our settings on Endnote and will be fixed.
Point 23, concerning L517: check this reference, is not complete yet.
Response 23: The list of references was updated.
Best wishes from all of the authors involved.
Reviewer 3 Report
Comments and Suggestions for Authors
The authors conducted a systematic review of studies focusing on sensor-based rumination detection in sheep, emphasizing the importance of rumination detection in monitoring the health of ruminants. Despite the abundance of studies on sensor-based rumination detection in cattle, there is a lack of studies pertaining to sheep. This study addresses this gap by introducing sensor-based rumination in sheep, making it acceptable with some minor revisions.
1. Table 1 and 2 are difficult to read. To enhance visualization, it is recommended to reduce the content.
1-1) In Table 1, please condense the 2nd (name/type), 3rd (system), 7th (position), 8th (classification method), and 9~12th (performance) columns. The 3rd column may not be necessary in this table and can be covered in the main text.
1-2) In Table 1, It is suggested that studies can be categorized based on sensor position.
1-3) Contents in Table 2 can also be reduced.
1-4) Please add titles of the Table 1 and 2.
2) In section 3.2, the utilization of deep learning approaches can enhance classification performance. For example, challenges arising from data scarcity or variations in devices/sensor positions can be addressed through transfer learning techniques. Here are some recommended references:
Mao, A.; Huang, E.; Gan, H.; Liu, K. FedAAR: A Novel Federated Learning Framework for Animal Activity Recognition with Wearable Sensors. Animals 2022, 12, 2142. https://doi.org/10.3390/ani12162142
Kleanthous N, Hussain A, Khan W, Sneddon J, Liatsis P. Deep transfer learning in sheep activity recognition using accelerometer data. Expert Systems with Applications. 2022 Nov 30;207:117925.
Ahn, S.-H.; Kim, S.; Jeong, D.-H. Unsupervised Domain Adaptation for Mitigating Sensor Variability and Interspecies Heterogeneity in Animal Activity Recognition. Animals 2023, 13, 3276. https://doi.org/10.3390/ani13203276
Author Response
Dear Sir or Madam,
We appreciate your review and valuable feedback on the article titled “Rumination Detection in Sheep: A Systematic Review of Sensor-Based Approaches.” The authors carefully considered each of your comments and have made the necessary revisions to address the concerns you raised. Below are our responses to your comments:
General comment: The authors conducted a systematic review of studies focusing on sensor-based rumination detection in sheep, emphasizing the importance of rumination detection in monitoring the health of ruminants. Despite the abundance of studies on sensor-based rumination detection in cattle, there is a lack of studies pertaining to sheep. This study addresses this gap by introducing sensor-based rumination in sheep, making it acceptable with some minor revisions.
Response to the general comment:
Thank you for this encouraging comment.
Point 1:
- Table 1 and 2 are difficult to read. To enhance visualization, it is recommended to reduce the content.
Response to Point 1:
The authors believe that the content presented here is important for understanding the best performances achieved for rumination detection and for putting them into perspective (i.e., which settings were used, and understanding they these results were achieved in an experimental farm with adult sheep). In order to clarify the necessity of this table, This passage was included in Section 3.1.3.: “Table 1 provides an overview of the best published performances regarding rumination detection. The table includes studies that assessed system performance, totaling 8 out of 17 studies. The percentages included present the highest percentages attained in the respective study, which is considered sufficiently representative according to an overall accuracy rating conducted by Barkved (2022). This assessment was further affirmed by Allwright (2022), who provided a specific accuracy score for machine learning models in their descriptions. This approach was chosen to provide a clear and concise summary of the best results achieved in different studies. Presenting a comprehensive overview of all results across various papers would have been far too many numbers variables to fit into one clear table in this publication, since many different settings and sensor positions were tested. So, the authors concentrated on the systems with the best performance. The table is arranged alphabetically according to the last names of the primary authors.” To enhance visualization, the table should be rotated 90° counterclockwise and expanded in width. The authors accidentally forgot to include the title, which should be: “An overview of sensor-based rumination detection in sheep: device types, testing environments, animals used, sensor positions and classification metrics.” To shorten the text in the table, the classification methods were listed as acronyms. The acronyms for the methods for classification should then be listed in a legend as follows: “Ca-nonical discriminant analysis (CDA), discriminant analysis (DA), machine learning (ML), Random Forest (RF), Support Vector Machine (SVM) andlinear discriminant analysis (LDA), k nearest neighbour (kNN) and adaptive boosting (Adaboost)”.
Point 2:
1-1) In Table 1, please condense the 2nd (name/type), 3rd (system), 7th (position), 8th (classification method), and 9~12th (performance) columns. The 3rd column may not be necessary in this table and can be covered in the main text.
Response to Point 2:
The authors believe that all of the information provided is essential for getting an overview of how the classification performances were achieved. Changing the layout of the table (meaning a counter clockwise rotation) would enhance the visualization.
Point 3:
1-2) In Table 1, It is suggested that studies can be categorized based on sensor position.
Response to Point 3:
Thank you for this suggestion. However, the studies cannot be categorized based on sensor position, seeing as some of the studies tested different sensor positions. The aim of this table is to show which settings and which sensor position yielded the best classification results for rumination detection.
Point 4:
1-3) Contents in Table 2 can also be reduced.
Response to Point 4:
The authors believe that the table presented here includes the necessary information to provide a comprehensive overview of how sensors which can detect rumination have been employed in scientific studies.
Point 5:
1-4) Please add titles of the Table 1 and 2.
Response to Point 5:
The proposed titles were added .
Point 6:
2) In section 3.2, the utilization of deep learning approaches can enhance classification performance. For example, challenges arising from data scarcity or variations in devices/sensor positions can be addressed through transfer learning techniques. Here are some recommended references:
Mao, A.; Huang, E.; Gan, H.; Liu, K. FedAAR: A Novel Federated Learning Framework for Animal Activity Recognition with Wearable Sensors. Animals 2022, 12, 2142. https://doi.org/10.3390/ani12162142
Kleanthous N, Hussain A, Khan W, Sneddon J, Liatsis P. Deep transfer learning in sheep activity recognition using accelerometer data. Expert Systems with Applications. 2022 Nov 30;207:117925.
Ahn, S.-H.; Kim, S.; Jeong, D.-H. Unsupervised Domain Adaptation for Mitigating Sensor Variability and Interspecies Heterogeneity in Animal Activity Recognition. Animals 2023, 13, 3276. https://doi.org/10.3390/ani13203276
Response to Point 6:
Thank you very much for this input and for sending us relevant publications. Since only one of the articles included in our search strategy addressed your point, we included this aspect as part of a limitation in the discussion. The following passage was included: “Some relevant aspects were not thoroughly discussed in the articles retrieved through this systematic review, such as how the use of deep learning approaches has the potential to improve classification performance. This was only addressed in one publication (Turner et al. [7]). For instance, challenges stemming from data scarcity or variations in sensor po-sitions can be effectively addressed through the application of transfer learning tech-niques, as discussed by Mao et al. [44], Kleanthous et al. [45] and Ahn et al. [46].”